# YeeD is an essential partner for YeeE-mediated thiosulfate uptake in bacteria and regulates thiosulfate ion decomposition

Mai Ikei[1]☯, Ryoji Miyazaki[1]☯, Keigo Monden[1], Yusuke Naito[1], Azusa Takeuchi[1], Yutaro S. Takahashi[1], Yoshiki Tanaka[1], Keina Murata[2], Takaharu Mori[2], Muneyoshi Ichikawa[3]*, Tomoya Tsukazaki[1]*

1 Division of Biological Science, Graduate School of Science and Technology, Nara Institute of Science and Technology, Ikoma, Nara, Japan, 2 Department of Chemistry, Faculty of Science, Tokyo University of Science, Shinjuku-ku, Tokyo, Japan, 3 State Key Laboratory of Genetic Engineering, Department of Biochemistry and Biophysics, School of Life Sciences, Fudan University, Shanghai, China

☯ These authors contributed equally to this work.
* ichikawa_muneyoshi@fudan.edu.cn (MI); ttsukaza@bs. naist.jp (TT)

**Data Availability Statement:** Coordinates and structure factors have been deposited in the

## Abstract

Uptake of thiosulfate ions as an inorganic sulfur source from the environment is important for bacterial sulfur assimilation. Recently, a selective thiosulfate uptake pathway involving a membrane protein YeeE (TsuA) in *Escherichia coli* was characterized. YeeE-like proteins are conserved in some bacteria, archaea, and eukaryotes. However, the precise function of YeeE, along with its potential partner protein in the thiosulfate ion uptake pathway, remained unclear. Here, we assessed selective thiosulfate transport via *Spirochaeta thermophila* YeeE in vitro and characterized *E. coli* YeeD (TsuB) as an adjacent and essential protein for YeeE-mediated thiosulfate uptake in vivo. We further showed that *S. thermophila* YeeD possesses thiosulfate decomposition activity and that a conserved cysteine in YeeD was modified to several forms in the presence of thiosulfate. Finally, the crystal structures of *S. thermophila* YeeE-YeeD fusion proteins at 3.34-Å and 2.60-Å resolutions revealed their interactions. The association was evaluated by a binding assay using purified *S. thermophila* YeeE and YeeD. Based on these results, a model of the sophisticated uptake of thiosulfate ions by YeeE and YeeD is proposed.

## Introduction

From bacteria to eukaryotes, sulfur is a vital element for cellular activities. For example, sulfur-containing biomolecules, such as L-cysteine, L-methionine, thiamine, glutathione, and biotin, play a variety of essential roles in cells [1]. Bacteria and plants can utilize L-cysteine as a source of sulfate, but they also have sulfur assimilation pathways to synthesize L-cysteine from inorganic sulfur compounds. In bacteria, L-cysteine is important not only as a component for protein synthesis but also as a reducing agent against oxidative stress [2]. There are 2 pathways for bacterial L-cysteine synthesis, which use O-acetylserine as a precursor, called the sulfate and

Protein Data Bank under accession number 8J4C and 8K1R. All data needed to evaluate the conclusions in the paper are present in the paper and/or the Supplementary Materials.

**Funding:** This work was supported by JSPS/MEXT KAKENHI (Grant No. JP21H05157 to T.M., Grant Nos. JP22K15075, JP20K15733 to Mu.I, Grant No. JP22K15061, JP22H05567 to R.M., and Grant Nos. JP22H02567, JP22H02586, JP21H05155, JP21H05153, JP21K19226, JP21KK0125 to T.T.), MEXT as "Program for Promoting Researches on the Supercomputer Fugaku" (Development and application of large-scale simulation-based inferences for biomolecules JPMXP1020230119) and HPCI project (hp200064 and hp230209) to T. M., PRESTO (JPMJPR20E1 to Mu.I) from the Japan Science and Technology Agency (JST), and private research foundations (the Chemo-Sero-Therapeutic Research Institute, Naito Foundation, Takeda Science Foundation, G-7 Scholarship Foundation, the Sumitomo Foundation, the Institute for Fermentation (Osaka), Yamada Science Foundation, KOSÉ Cosmetology Research Foundation, and Japan Foundation for Applied Enzymology) to T.T. The funders had no role in study design, data collection and analysis, decision to publish, or preparation of the manuscript.

**Competing interests:** The authors have declared that no competing interests exist.

**Abbreviations:** BLI, biolayer interferometry; DDM, dodecyl β-maltoside; DTT, dithiothreitol; IAA, iodoacetamide; IPTG, isopropyl β-D-thiogalactopyranoside; LCP, lipidic cubic phase; MALDI-TOF, matrix-assisted laser desorption ionization–time of flight; MD, molecular dynamics; MS, mass spectrometry; NEM, N-Ethylmaleimide; PMSF, phenylmethylsulfonyl fluoride; SA, sinapinic acid; SSM, solid-supported membrane.

thiosulfate pathways [3]. In the sulfate pathway, sulfate ion is first decomposed into sulfide ion through phosphorylation by 2 molecules of ATP and reduction by 4 molecules of NADPH. Subsequently, L-cysteine is synthesized from sulfide ion and O-acetylserine by O-acetylserine sulfhydrylase-A (CysK). In the thiosulfate pathway, S-sulfocysteine is synthesized from thiosulfate ion and O-acetylserine by O-acetylserine sulfhydrylase-B (CysM); S-sulfocysteine is then reduced by 1 NADPH molecule, and L-cysteine is synthesized. The sulfate and thiosulfate ions used in these pathways are taken up from the environment by transporters on the cytoplasmic membrane. Sulfate and thiosulfate ions are trapped by periplasmic proteins Sbp [4] and CysP [5], respectively. Both sulfate and thiosulfate ions are then passed on to a complex formed by the inner membrane proteins CysU and CysW and the cytosolic protein CysA (CysUWA) (also called CysTWA) [3] and transported into the cytoplasm by the CysUWA complex using the energy from ATP hydrolysis.

A bacterial membrane protein, YeeE, has been identified as mediating thiosulfate uptake based on growth complementation assay of *Escherichia coli* cells, and the structure of *Spirochaeta thermophila* YeeE (*St*YeeE) has been determined by X-ray crystallography [6]. YeeE was proposed to be almost entirely buried in the membrane and shows a unique hourglass-like structure. An electron density near a conserved cysteine residue on the outside surface was assigned to a thiosulfate ion, which seemed to be connected to the sulfur atom of the cysteine via a S─H─S hydrogen bond [6]. In addition, the binding of thiosulfate ions to purified YeeE protein was shown by isothermal titration calorimetry experiments. In the center of YeeE, 3 conserved, functionally important cysteines, including the thiosulfate-interacting one, are arranged side by side and perpendicular to the membrane. Based on these structural features and in vivo functional analysis using a series of mutants, thiosulfate ion was proposed to be transported from the environment to the cytoplasm while transiently interacting with the 3 conserved cysteine residues by relays mediated via hydrogen bonds [6], independently of the thiosulfate pathway described above. However, it has not been elucidated whether YeeE alone transports thiosulfate ions. Moreover, no YeeE-associated protein has been clearly defined, although 1 candidate gene is *yeeD*, which resides within the same operon that includes *yeeE* [7], and the regulatory mechanism of the alternative, sophisticated thiosulfate uptake by YeeE remains unclear.

Recently, in addition to the *E. coli yeeE* gene, *yeeD* was shown to be involved in t̲h̲i̲o̲s̲u̲l̲f̲a̲t̲e̲ u̲p̲t̲a̲k̲e̲ and the two were named *tsuA* and *tsuB*, respectively [8]. YeeD is a cytoplasmic protein that belongs to the TusA (tRNA 2-t̲h̲i̲o̲u̲r̲i̲d̲i̲n̲e̲ s̲y̲n̲t̲h̲e̲s̲i̲z̲i̲n̲g̲ protein A) protein family [9]. TusA plays various roles in sulfate transfer activities in cells, such as thiomodification of tRNA [10], molybdenum cofactor biosynthesis [11], and dissimilatory sulfur and tetrathionate oxidation [12]. The cysteine residue in a Cys-Pro-X-Pro (CPxP) motif of TusA receives activated sulfur from the L-cysteine desulfurase IscS [13]. Although YeeD possesses the CPxP motif, it cannot complement TusA function [11]. Therefore, YeeD is thought to have a sulfate-related yet distinct function from TusA in bacteria. In some bacteria, such as gram-positive *Corynebacterium* species YeeE and YeeD are encoded as 1 polypeptide, implying that they function together in the thiosulfate uptake pathway. However, the enzymatic activity of YeeD and the functional cooperativity of YeeE and YeeD have not been well characterized.

In this study, first, we measured thiosulfate uptake activity using *St*YeeE-reconstituted liposomes to clarify YeeE function. Second, according to an *E. coli* growth complementation assay, we found that *E. coli* YeeD (*Ec*YeeD) plays an essential role in YeeE function in vivo. Third, we demonstrated a thiosulfate decomposition activity in purified *S. thermophila* YeeD (*St*YeeD). Fourth, direct interaction between *St*YeeE and *St*YeeD proteins was detected, and substrate thiosulfate ions weakened the binding of these proteins. Fifth, the crystal structures of *St*YeeE–YeeD complex were determined at 3.34 Å and 2.60 Å resolutions. Critical residues of YeeD for

both its activity and interaction with YeeE were defined. Based on these results, detailed mechanisms for the functional cooperation between YeeE and YeeD in thiosulfate uptake are discussed.

## Results

### YeeE-mediated thiosulfate transport

Although *E. coli* growth complementation tests showed that YeeE is involved in the uptake of thiosulfate ions [6], thiosulfate transport via YeeE has not yet been demonstrated. To detect thiosulfate uptake activity using a purified reconstituted system, we adapted solid-supported membrane (SSM)-based electrophysiology [14]. First, *St*YeeE-reconstituted liposomes were applied to the SSM, and charge displacement was measured in the presence of sulfate ions or thiosulfate ions (Fig 1A). A change in the potential toward the negative direction was observed in the presence of thiosulfate ions but not sulfate ions, meaning that negatively charged thiosulfate ions were selectively transported into the proteoliposomes. The slight change in the potential toward the positive direction in the presence of sulfate ions was probably due to buffer shock since there was no counter ion present in these measurements. Next, because *St*YeeE-reconstituted liposomes showed no transport activity for sulfate ions, we measured YeeE activity using sulfate as a counter ion. As a result, a clear peak toward the negative direction ($-0.62$ nA) was observed only in the presence of *St*YeeE (Fig 1B). The negative peak area increased when higher concentrations of thiosulfate ions were used (Fig 1C). These experiments verified that YeeE transports thiosulfate ions specifically but not sulfate ions.

### YeeD is essential for the YeeE-mediated thiosulfate pathway

According to recent reports [7,8], YeeD may function in combination with YeeE in *E. coli*. YeeD is composed of about 80 amino acid residues and possesses 1 or 2 cysteines, which are highly conserved among bacteria. In some organisms, such as *E. coli*, YeeD has 2 conserved cysteine residues (C13 and C39 in *Ec*YeeD), while other organisms like *S. thermophila* have 1 conserved cysteine residue (C17 in *St*YeeD) (Fig 2A). The invariant cysteine residue in the N-terminal region is part of a CPxP motif that is conserved among TusA family proteins (Figs 2A and S1). The predicted *Ec*YeeD structure in the AlphaFold2 database [15,16] shows a globular

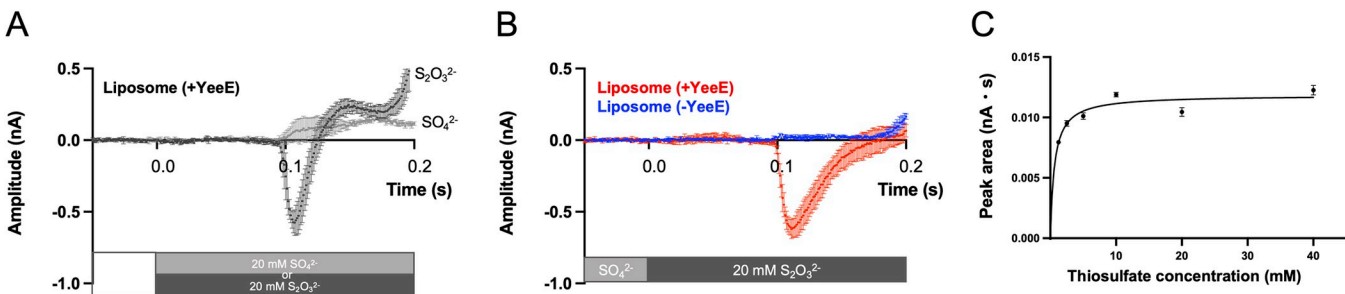

**Fig 1. Thiosulfate ion transport activity of *St*YeeE-reconstituted liposomes.** (**A**) Selective uptake of thiosulfate ion by *St*YeeE. *St*YeeE-reconstituted liposomes were used for the measurement using the SSM method. The nonactivation buffer, lacking any sulfur-related ions, was exchanged with buffer containing either 20 mM $Na_2S_2O_3$ or 20 mM $Na_2SO_4$ at 0.0 s. (**B**) YeeE-dependent negative current changes. Liposomes with and without *St*YeeE were used. The nonactivation buffer containing 20 mM $Na_2SO_4$ was exchanged with buffer containing 20 mM $Na_2S_2O_3$ instead of $Na_2SO_4$ at 0.0 s. The current values from 3 measurements were standardized based on the average values from $-0.05$ to 0.0 s, and then the mean values from the 3 measurements were calculated. Error bars indicate the SD (standard deviation). (**C**) Thiosulfate concentration dependency of the uptake activity. The current changes were monitored in the same manner as in (B), except for the concentration of thiosulfate and sulfate ions (1.25, 2.5, 5, 10, 20, and 40 mM). The negative peak area values (nA·s) from 3 measurements were calculated and plotted with the SD. The plot was fitted with Michaelis–Menten equation, and $V_{max}$ was estimated as 0.01184 nA·s and $K_m$ as 0.6261 mM. The underlying data can be found in S1 Data.

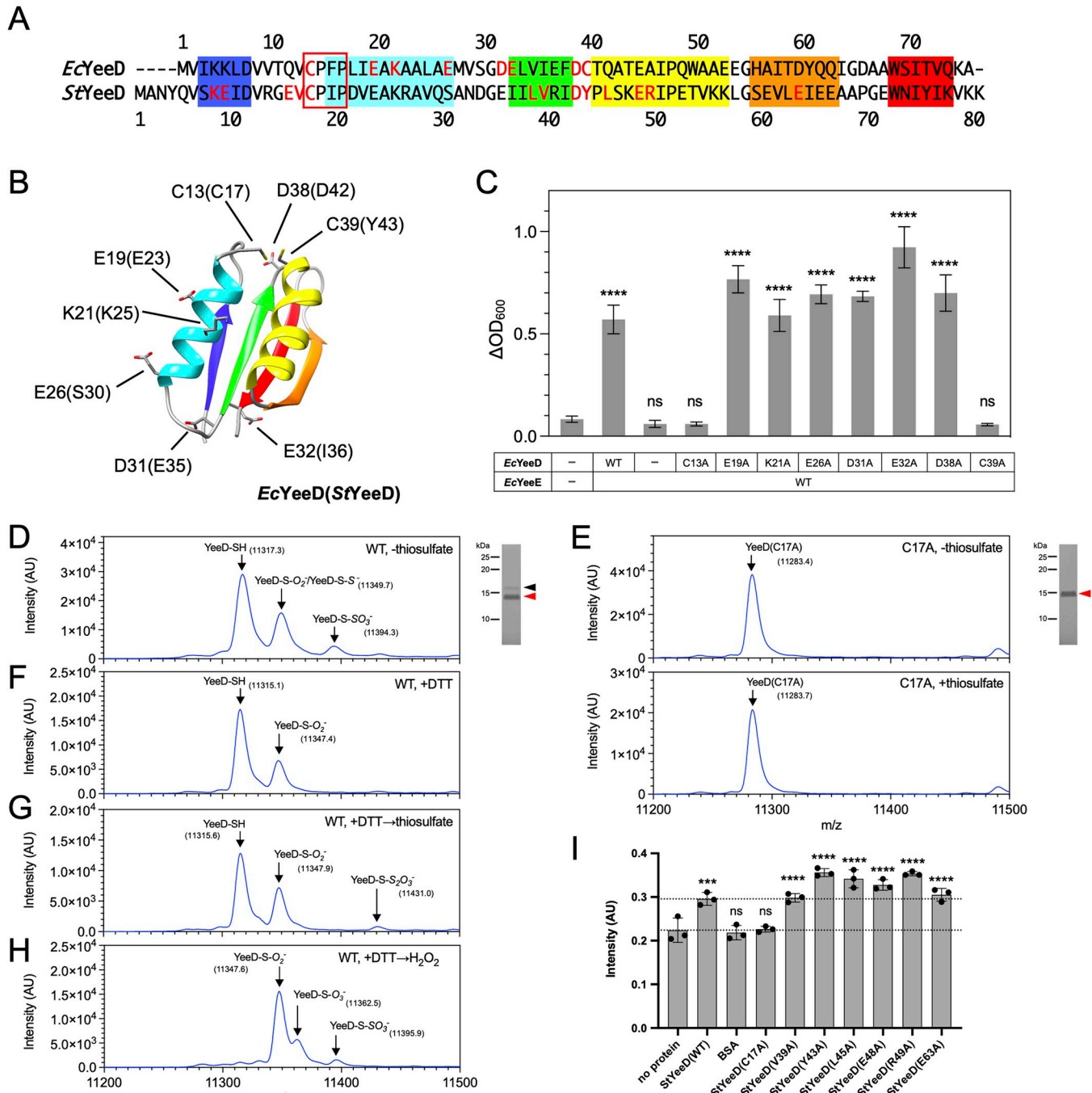

**Fig 2. Characterization of YeeD functions in vivo and in vitro.** (**A**) Sequence alignment of *Ec*YeeD and *St*YeeD. The regions of α-helices and β-strands predicted by AlphaFold2 in (B) are highlighted with colors. Red text indicates amino acid residues that were mutated in this study. The conserved CPxP motif is enclosed in a red rectangle. (**B**) Cartoon model of AlphaFold2-predicted structure of *Ec*YeeD. α-Helices and β-strands are colored as in (A). The side chains of amino acid residues mutated for growth complementation analysis in (C) are shown as stick models. Corresponding amino acid residues from *St*YeeD are also indicated in parentheses. (**C**) Growth complementation assay of Δ*cysPUWA* Δ*yeeE* (DE3) cells, depending on *Ec*YeeE and *Ec*YeeD expression from plasmids. The mean values of increased OD$_{600}$ (ΔOD$_{600}$) after 24 h are shown. Error bars represent the SD from 3 measurements. Statistical significance compared with the Δ*cysPUWA* Δ*yeeE* (DE3) cells possessing an empty vector was determined using one-way analysis of variance (ANOVA) followed by Dunnett's multiple comparisons tests (****, $p < 0.0001$; ns, not significant). (**D**) CBB-stained SDS-PAGE gel and mass spectrometry of purified *St*YeeD(WT). Red and black arrowheads indicate the major and minor bands of purified WT *St*YeeD, respectively. In mass spectrometry, 3 major peaks corresponding to -SH, -S-$O_2^-$/-S-S$^-$, and -S-$SO_3^-$ were detected. (**E**) CBB-stained SDS-PAGE gel and mass spectrometry of purified *St*YeeD(C17A) (upper) and 500 μM thiosulfate-treated *St*YeeD(C17A) (lower). In mass spectrometry, a single peak was observed. (**F**) Mass spectrometry of DTT-reduced *St*YeeD(WT). (**G**) Mass spectrometry of 500 μM thiosulfate-treated *St*YeeD after DTT-reduction. (**H**) Mass spectrometry of *St*YeeD oxidized by hydrogen peroxide after DTT-

reduction. (**I**) Measurement of in vitro enzymatic activity of *St*YeeD. Fluorescence intensity changes derived from an H$_2$S-detectable fluorescent probe, HSip-1, in the presence of thiosulfate (500 μM) were measured using a series of purified *St*YeeD derivatives in 10 mM Tris-HCl (pH 8.0), 300 mM NaCl, and 70 μM β-ME. Mean values of 3 experiments are shown. Error bars represent SD. Statistical significance compared with samples containing no protein was determined using one-way ANOVA followed by Dunnett's multiple comparisons tests (***, $p < 0.001$; ****, $p < 0.0001$; ns, not significant). Dashed lines indicate the mean values of no-protein samples and *St*YeeD(WT). The underlying data can be found in S1 Data. The original gel images of (D and E) can be found in S1 Raw Images.

shape with 2 α-helices and a β-sheet composed of 4 β-strands (Fig 2B), similar to other TusA protein structures [17,18]. To evaluate the influence of YeeD on YeeE activity in cells, a growth complementation assay using *E. coli* MG1655 *ΔcysPUWA ΔyeeE::kan* (DE3) cells was performed, as described previously [6]. These *E. coli* cells, which lack sulfate and thiosulfate ion-uptake pathways, cannot grow in the presence of thiosulfate as a sole sulfur source (Figs 2C, S2A, and S2B) but can when they carry a plasmid containing the *E. coli yeeE* operon, which includes *yeeE* and *yeeD* [6] (Fig 2C). When the *yeeD* region was deleted, growth was similar to the negative control, which was conducted using a vector without *yeeE* and *yeeD* regions (Fig 2C), revealing that YeeD is essential for thiosulfate uptake via YeeE in vivo. Further, we prepared a series of alanine mutations in *Ec*YeeD, including the conserved cysteine mutations C13A and C39A, that are located on the surface of the AlphaFold2 model (Fig 2B). Subsequent growth complementation assays using these YeeD mutations demonstrated the significance of C13 and C39, but not of the other mutated residues (E19, K21, E26, D31, E32, and D38), in *Ec*YeeD function (Fig 2C).

## Thiosulfate decomposition activity of YeeD and its catalytic center residue

To unveil the enzymatic function of YeeD itself, we purified *St*YeeD and its mutant proteins and then conducted their biochemical analyses in vitro. The gel filtration profile of purified His-tagged *St*YeeD(WT) (11.3 kDa) appears to be a mixture of dimers and monomers as compared to gel filtration markers (S2C Fig), similar to TusA [12]. Nonreducing and reducing SDS-PAGE after the addition of iodoacetamide (IAA) or N-Ethylmaleimide (NEM), which block the sidechain of Cys residues, showed that almost all *St*YeeD(WT) lacks disulfide bond formation: The dimeric *St*YeeD apparent in gel filtration profiles is not due to disulfide bond formation (S2D Fig). In the range of 10 to 15 kDa on SDS-PAGE gels, purified *St*YeeD(WT) exhibited main and faint minor bands, whereas *St*YeeD(C17A), a putative catalytic center mutant, exhibited only the main band (Fig 2D and 2E). Since this slightly upshifted band did not disappear even in reducing SDS-PAGE (S2D Fig), the -SH group might be modified to an irreversible oxidation state. Indeed, oxidation of WT *St*YeeD with hydrogen peroxide significantly increased the reductant-resistant band (S2E Fig), presumably due to irreversible formation of sulfinic acid (-SO$_2^-$) or sulfonic acid (-SO$_3^-$) [19]. To examine *St*YeeD modification at C17, we performed matrix-assisted laser desorption ionization-time of flight (MALDI-TOF) mass spectrometry (MS) of YeeD in several conditions. The samples for MS were prepared under aerobic conditions. The spectrum of purified YeeE in the absence of thiosulfate revealed 3 major peaks with m/z of 11,317.3, 11,349.7, and 11,394.3 (Fig 2D). The first peak (11,317.3 m/z) was assigned as unmodified *St*YeeD (-SH at C17) as its theoretical m/z value is 11,315.7. The second peak (11,349.7 m/z) might correspond to sulfinic acid (-S-$O_2^-$ at C17) and/or persulfide modification (-S-S$^-$ at C17) as it increased by 32.4 (close to the value of 2 oxygen atoms, 32.0, or 1 sulfur atom, 32.1). Although the formation of sulfinic acid (see above) may be an artifact due to oxidation in the purification process, the persulfide modification was also found by MS in purified *Pseudomonas aeruginosa* PA1006, an ortholog of *E. coli* TusA [20], which has one conserved cysteine residue corresponding to C17 of *St*YeeD. The third 77.0 m/z-increased peak (11,394.3 m/z) may be attributable to perthiosulfonic acid (-S-$SO_3^-$ at C17). In

contrast, $St$YeeD(C17A) showed only one prominent peak (11,283.4 m/z) as its theoretical unmodified value (11,283.6 m/z) (Fig 2E). These results suggest that the C17 of $St$YeeD is highly reactive to sulfur-related molecules and susceptible to sulfur($S_\gamma$)-related modifications. Next, we performed MS analysis of the $St$YeeD protein in the presence of DL-dithiothreitol (DTT), a reductant. The third peak entirely disappeared after the addition of DTT (Fig 2F). By contrast, the second peak remained even in the presence of DTT. Persulfide modification (-S-$S^-$) can be reduced by DTT [20], but sulfinic acid (-S-$O_2^-$), an overoxidation product, is irreversible as previously shown [19,21]. Therefore, in the DTT condition, the second peak is thought to be residual sulfinic acid product. Using sample treated with DTT, to gain insights into catalytic intermediates, MS analysis was performed in the presence of thiosulfate (Fig 2G). In the condition with thiosulfate (500 μM), a peak with m/z value of 11,431.0 appeared. The peak with an m/z value of 11,431.0 may correspond to -S-$S_2O_3^-$ because the m/z increments were equivalent to the addition of $S_2O_3$ to StYeeD(-SH). The previously reported crystal structure of TsdBA (PDB ID 5LO9) included the same modification of a cysteine residue's thiol (-SH) group to -S-$S_2O_3^-$, indicating that this modification is one of the stable forms [22]. In the case of StYeeD(C17A), the addition of thiosulfate did not increase the m/z (Fig 2E). In addition, we performed MS using $St$YeeD sample treated with DTT and then hydrogen peroxide (Fig 2H). In this condition, the peak corresponding to $St$YeeD(-SH) completely disappeared, but peaks corresponding to sulfinic acid (-S-$O_2^-$), sulfonic acid (-S-$O_3^-$), and perthiosulfonic acid (-S-$SO_3^-$) were observed. Given these results, although perthiosulfonic acid (-S-$SO_3^-$) detected in the purified StYeeD is thought to be due to oxidization during the purification procedure, $St$YeeD may temporarily take the -S-$SO_3^-$ form in the process of thiosulfate decomposition.

The additional binding of $S_2O_3$ to the C17 side chain, in form of -S-$S_2O_3^-$, and the existence of persulfide modification (-S-$S^-$ at C17) and perthiosulfonic acid (-S-$SO_3^-$ at C17), as indicated by the MS results, raise the possibility that YeeD decomposes thiosulfate ion ($S_2O_3^{2-}$) and releases sulfide ion ($S^{2-}$). To test this, we monitored hydrogen sulfide ($H_2S$) derived from thiosulfate-decomposed sulfide ions using a fluorescent probe for $H_2S$, HSip-1. A significant increase in fluorescence intensity was observed in the presence of $St$YeeD compared with an irrelevant protein, BSA, and with no protein (Figs 2I and S2G), suggesting that YeeD catalyzes the decomposition of thiosulfate as a substrate. We also tested the thiosulfate decomposition activity of several $St$YeeD mutants. Among these, only C17A did not show a significant increase in fluorescence; NEM-modified $St$YeeD also did not. These results indicated that the conserved cysteine residue of YeeD (C17 in $St$YeeD) is the catalytic center residue for thiosulfate decomposition.

## Crystal structure of the YeeE-YeeD complex

We succeeded in crystallizing a fusion protein of $St$YeeE and $St$YeeD, designed based on the amino acid sequence of *Corynebacterium pollutisoli* YeeED, which is expressed as a single protein. First, because a comparison of $St$YeeE/D and several YeeEDs' amino acid sequences revealed that *C. pollutisoli* YeeED possesses the shortest linker, of 40 amino acids, between its YeeE and YeeD regions, we prepared the fusion protein by introducing the 40 residues between the C-terminus of $St$YeeE and the N-terminus of $St$YeeD. Second, after attempting but failing to crystallize the fusion protein, $St$YeeE-$St$YeeD, we tried crystallization using alanine mutants of the conserved cysteines in $St$YeeE-$St$YeeD and succeeded in crystallizing $St$YeeE(C22A)-$St$YeeD. Using the X-ray diffraction dataset of $St$YeeE(C22A)-$St$YeeD, we determined the crystal structure of the $St$YeeE-$St$YeeD complex at 3.34 Å (S3A and S3B Fig, Table 1 8J4C). However, the resolution was insufficient to reveal the details of the interactions

between YeeE and YeeD. Introducing a mutation in YeeD (L45A) strengthened the interaction, as found in binding experiments using purified YeeE and YeeD mutants (described later). The diffraction dataset of crystals of *St*YeeE(C22A)-*St*YeeD(L45A) was improved to 2.60 Å (Fig 3A and Table 1 8K1R), and the resulting 2.60-Å crystal structure enabled us to discuss the detailed interactions between YeeE and YeeD.

**Table 1. Data collection and refinement.**

| PDB ID | 8J4C | 8K1R |
|---|---|---|
| Crystal | *St*YeeE(C22A)-*St*YeeD | *St*YeeE(C22A)-*St*YeeD(L45A) |
| **Data Collection** | | |
| X-ray source | SPring-8 BL32XU | SPring-8 BL32XU |
| Wavelength (Å) | 1.00 | 1.00 |
| Space group | $P2_122_1$ | $P2_122_1$ |
| $a$, $b$, $c$ (Å) | 73.98, 102.44, 186.52 | 73.58, 101.75, 182.76 |
| Resolution range (Å) | 47.60–3.34 (3.46–3.34) | 49.01–2.60 (2.69–2.60) |
| Total reflections | 432,750 (38,159) | 13,803,448 (1,256,177) |
| Unique reflections | 21,249 (2,011) | 42,977 (4,206) |
| Multiplicity | 20.4 (18.4) | 321.2 (297.1) |
| Completeness (%) | 99.04 (96.87) | 99.52 (99.32) |
| $I/\sigma(I)$ | 4.12 (0.83) | 15.25 (2.10) |
| Wilson B-factor | 49.19 | 31.95 |
| R-pim | 0.2207 (1.017) | 0.0478 (0.2706) |
| $CC_{1/2}$ | 0.963 (0.578) | 0.996 (0.861) |
| | | |
| **Refinement** | | |
| Reflections used in refinement | 21,070 (2,011) | 42,846 (4,206) |
| Reflections used for R-free | 1,990 (190) | 1,993 (196) |
| $R_{work}$ | 0.2654 (0.3596) | 0.2064 (0.2705) |
| $R_{free}$ | 0.3095 (0.4113) | 0.2547 (0.3108) |
| Number of nonhydrogen atoms | 6,675 | 6,900 |
| protein | 6,238 | 6,247 |
| thiosulfate | 10 | 0 |
| monoolein | 427 | 545 |
| water | 0 | 108 |
| RMS derivations bond length (Å) | 0.003 | 0.009 |
| bond angles (°) | 0.56 | 1.16 |
| MolProbity Validation overall score | 1.95 | 1.80 |
| clashscore | 11.91 | 7.67 |
| rotamer outliers (%) | 0.00 | 1.72 |
| Ramachandran favoured (%) | 94.80 | 96.77 |
| allowed (%) | 4.95 | 2.73 |
| outliers (%) | 0.25 | 0.50 |
| Average B-factor | 66.46 | 39.54 |
| protein | 67.97 | 38.61 |
| thiosulfate | 73.25 | - |
| monoolein | 44.20 | 51.12 |
| water | - | 34.03 |

Statistics for the highest-resolution shell are shown in parentheses.

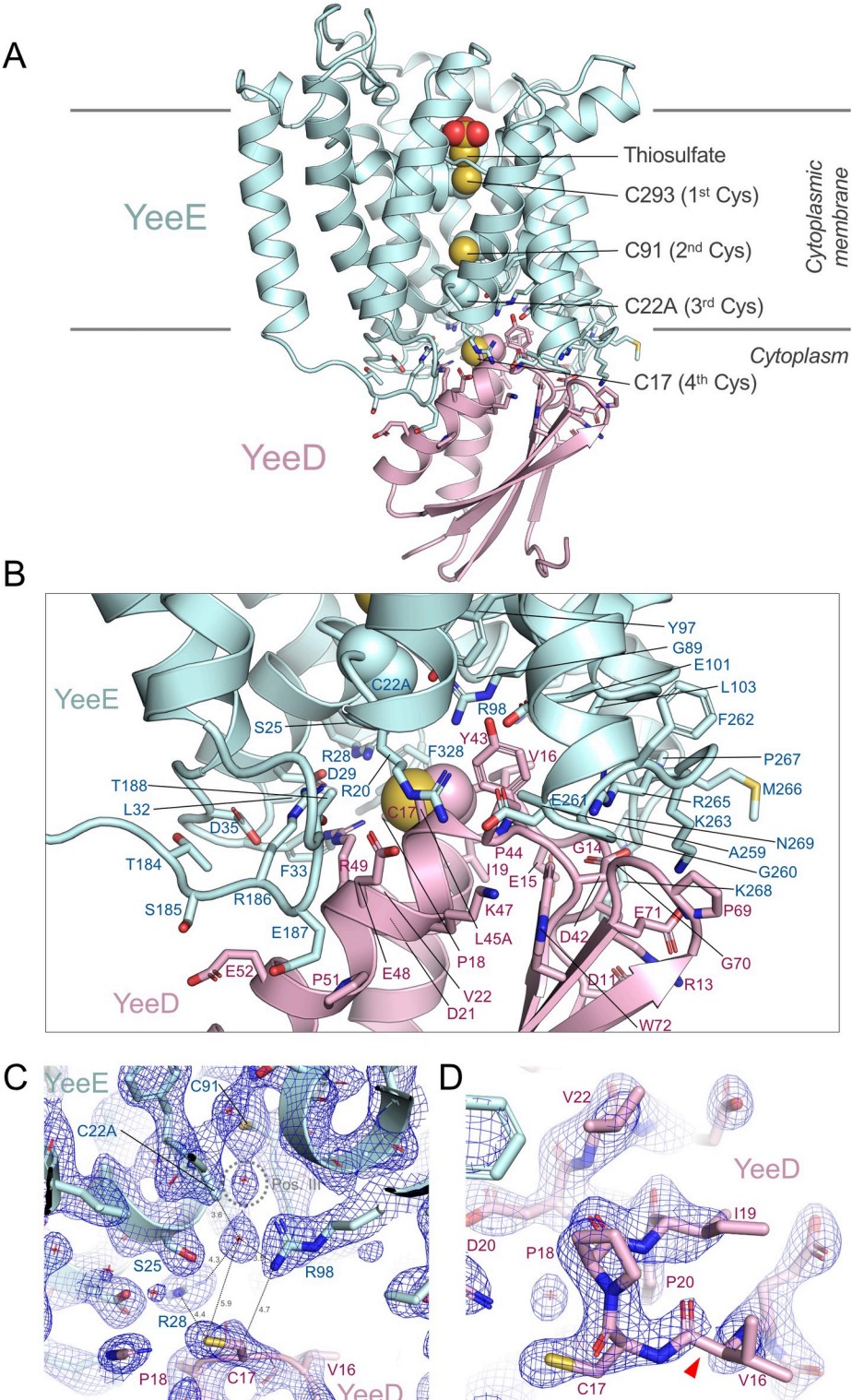

**Fig 3. Crystal structure of *St*YeeE-*St*YeeD complex (PDB ID 8K1R).** (**A**) Cartoon model of *St*YeeE(C22A)-YeeD (L45A). The side chains of amino acid residues on the contact surfaces between *St*YeeE and *St*YeeD are shown as stick models. The side chains of the conserved cysteine positions are shown in the spheres model. The thiosulfate ion in the *St*YeeE(C22A)-YeeD model (PDB ID 8J4C) was superimposed as spheres model. (**B**) Close-up view of the contact region. (**C**) 2Fo-Fc electron density map at 1.5 δ of the border region around C22A in *St*YeeE and C17 in *St*YeeD. Position III is a previously proposed thiosulfate-binding site. The red stars indicate water molecules. (**D**) 2Fo-Fc electron density map at 1.3 δ around the CPxP motif. The red arrowhead indicates where electron density map is faint.

Each crystal lattice in the datasets was the same, except for having slightly different unit cell dimensions (Table 1). The asymmetric unit contains 2 molecules of *St*YeeED, Mol A and Mol B. In the crystal packing, *St*YeeD in Mol A is involved in interacting with an adjacent YeeD region, but YeeD in Mol B is not (S3B Fig). When the 2 *St*YeeE-*St*YeeD structures (Mol A and Mol B), showing different packing interactions, were superimposed, the crystal structures were almost identical with an RMSD of 0.433 Å for 8J4C or 0.155 Å for 8K1R for Cα atoms (S3A Fig). Although the orientation of YeeD relative to YeeE is slightly different, the RMSD between 8J4C and 8K1R for Cα atoms is 0.896 (S3C Fig) and the active site of YeeD is similar. Therefore, in this paper, we introduce Mol A of the higher resolution structure, which shows the best electron density map, to explain YeeE and YeeD interactions.

The overall structure of YeeE in the crystal structure of *St*YeeE-*St*YeeD resembles the previously reported crystal structure of *St*YeeE (PDB ID 6LEP) [6], with an RMSD of 0.596 Å for Cα atoms (S3D Fig). The cytoplasmic indentation of *St*YeeE associates with *St*YeeD, which consists of 2 α-helices and 4 β-sheets. The architecture of *St*YeeD is similar in shape to the sequence-homologous *E. coli* TusA (22.5% identical) (PDB ID 3LVK) (S3E Fig) [18]. Residues involved in the YeeE–YeeD interaction are shown in Fig 3B. C22 of *St*YeeE is located at the interface, implying that C22A affects the interaction. Residues R28, R98, E101, and E261 on the cytoplasmic surface, conserved in YeeEs [6], were found to be involved in the interaction with YeeD. While the important, conserved residues C293, C91, and C22, termed the first, second, and third cysteines, respectively, are arranged perpendicular to the membrane at 8.6- and 6.7-Å intervals, C17 of *St*YeeD, one of the YeeE-interacting residues, is the putative enzymatic active site and is located adjacent to C22 of *St*YeeE as the "fourth cysteine." This positioning means that a thiosulfate ion transported through the membrane via the center region of YeeE interacts with the first-to-third cysteines of YeeE in succession and then interacts with the fourth cysteine of YeeD. The distance between the third and the fourth cysteines, 10.8 Å, is wider than that between the other cysteines, and this space may be needed for transient binding of the thiosulfate ion rather than its transport (Fig 3C). A water molecule is located at position III, a predicted thiosulfate-binding site [6]. A position where another water molecule is located below position III corresponds to the position of the side chain of C22 [6]. The relay of a thiosulfate ion from position III to C17 is likely facilitated by the conserved positively charged residues R28 and R98 (Fig 3C). The CPxP motif forms a *cis* peptide between C17 and P18 and is located at the beginning of the α-helix (Fig 3D), similar to a previous report [17]. In addition, the electron density of the peptide bonds between V16 and C17 is faint, which may be closely related to C17 being the active center.

## Interaction between YeeE and YeeD

To confirm that the interactions between YeeE and YeeD revealed in the crystal structures of *St*YeeE-YeeD fusion protein also occur in solution, we performed a binding assay based on the biolayer interferometry (BLI) method [23] using purified *St*YeeE and His-tagged *St*YeeD. Clear association and dissociation of *St*YeeE with an *St*YeeD-solidified Ni-NTA sensor were observed as real-time wavelength shifts (Fig 4A, blue). Based on the *St*YeeE binding-dependent wavelength changes, the $K_D$ between *St*YeeD and *St*YeeE was estimated as 1.6 ± 0.20 μM. When a buffer containing sulfate ion was used, the binding was similar to that in a buffer without a sulfur source (S4A and S4B Fig). In contrast, *St*YeeE showed significantly decreased binding to *St*YeeD in a buffer containing thiosulfate ions (S4A and S4B Fig). We next sought the important residues for the interaction using a series of *St*YeeD mutants (S2F Fig). The mutants E15A, V16A, C17A, D42A, Y43A, E48A, and R49A severely impaired *St*YeeD's

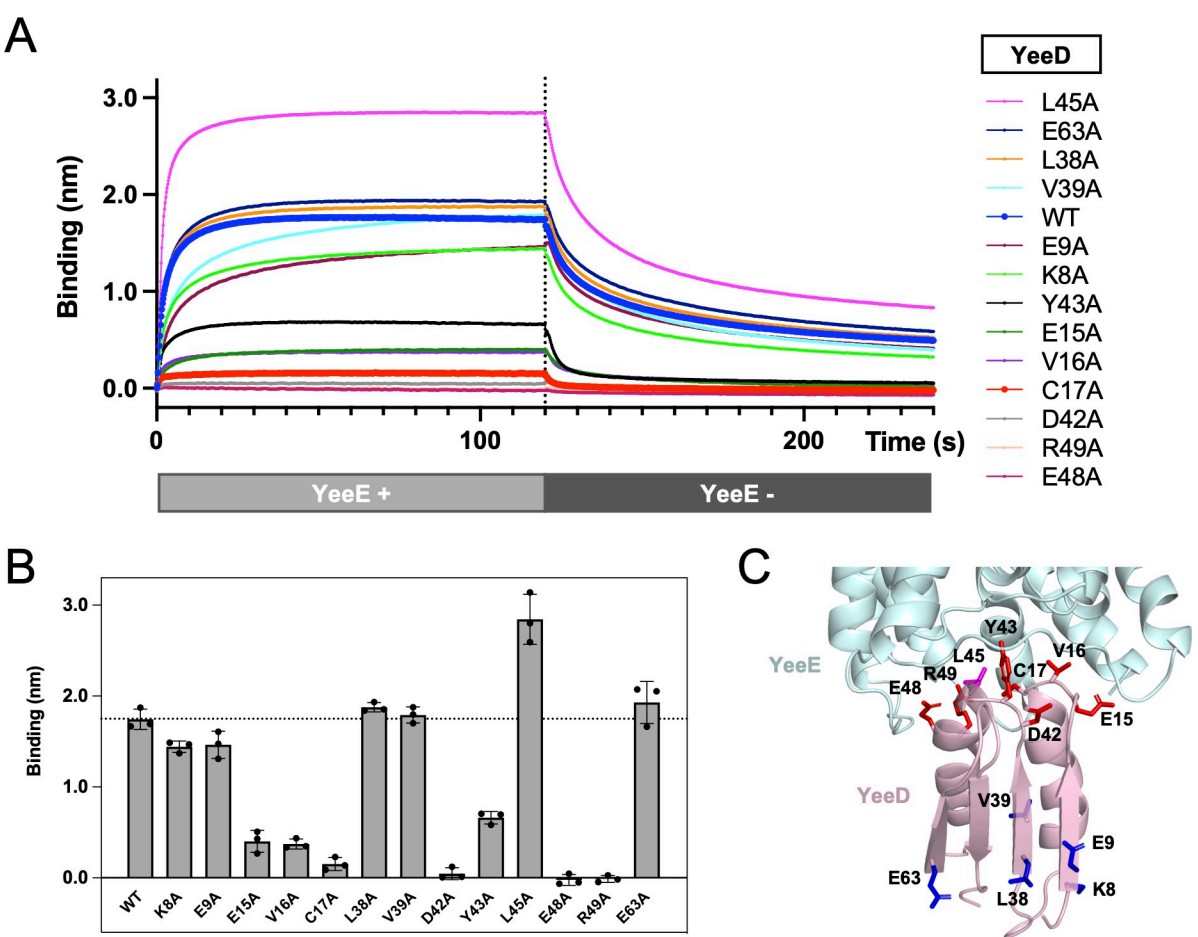

**Fig 4. Interaction between *St*YeeE and *St*YeeD.** (**A**) Real-time detection by the BLI method of association and dissociation of *St*YeeE with/from solidified *St*YeeD and its derivatives with Octet buffer (20 mM Tris-HCl (pH 8.0), 300 mM NaCl, 0.1% (w/v) DDM, 1 mM β-ME). Each line shows the mean value of 3 measurements. The dashed line indicates the 120-s point where *St*YeeD-solidified biosensors were submerged in the YeeE-free buffer. (**B**) Mean values of *St*YeeE binding to solidified *St*YeeD at 120 s. Error bars show the SD of 3 measurements. The dashed line indicates the mean value of *St*YeeD(WT). (**C**) Mutation site mapping on the crystal structure of *St*YeeE-YeeD complex (PDB ID 8J4C). The side chains of mutated amino acid residues in (A) are shown as stick models. Magenta, blue, and red residues indicate that those Ala-substituted mutants showed increased, similar, and reduced binding affinity, respectively, compared to WT. The underlying data can be found in S1 Data.

affinity for *St*YeeE, while the mutant L45A enhanced the affinity (Fig 4B). This enhanced affinity could stabilize the crystal packing of *St*YeeE-*St*YeeD, leading to its higher-resolution structure. In the crystal structure, most of these influential residues for affinity were located at the border region between YeeE and YeeD (Fig 4C). On the other hand, Ala substitutions on the opposite side maintained affinity (Fig 4). The calculated $K_a$, $K_d$, and $K_D$ values for YeeD mutant proteins are summarized in Table 2. Notably, the conserved cysteine residue (C17) of *St*YeeD proved to be pivotal for not only the enzyme activity but also the interaction with *St*YeeE. These BLI results in solution are consistent with the crystal structure.

To further assess the important residues of YeeE for the interaction with YeeD, we used a growth complementation assay. *Ec*YeeE mutations of residues at the interaction surface (R21A, R101A, E104A, E271A, and R275A) could not complement the growth of *E. coli* MG1655 Δ*cysPUWA* Δ*yeeE*::*kan* (DE3) cells (S4C and S4D Fig). These results emphasize that the interaction of YeeE and YeeD is essential for in vivo function of thiosulfate uptake.

**Table 2. $k_a$, $k_d$, and $K_D$ values of the interactions between YeeE and YeeD proteins.**

| | $k_a$ (1/μM·s) | $k_d$ (1/s) | $K_D$ (μM) |
|---|---|---|---|
| WT | 0.017 ± 0.0029 | 0.027 ± 0.0029 | 1.6 ± 0.20 |
| K8A | 0.0089 ± 0.0011 | 0.037 ± 0.0019 | 4.3 ± 0.61 |
| E9A | 0.0060 ± 0.00058 | 0.027 ± 0.0021 | 4.6 ± 0.22 |
| E15A | 0.0075 ± 0.0013 | 0.071 ± 0.033 | 9.5 ± 4.2 |
| V16A | 0.017 ± 0.0016 | 0.093 ± 0.023 | 5.6 ± 0.86 |
| C17A | * | * | * |
| L38A | 0.018 ± 0.0013 | 0.032 ± 0.0030 | 1.7 ± 0.055 |
| V39A | 0.0058 ± 0.00033 | 0.032 ± 0.00058 | 5.4 ± 0.30 |
| D42A | ** | ** | ** |
| Y43A | 0.0082 ± 0.00095 | 0.16 ± 0.017 | 20 ± 2.1 |
| L45A | 0.029 ± 0.0016 | 0.035 ± 0.0040 | 1.2 ± 0.10 |
| E48A | 0.017 ± 0.010 | 0.14 ± 0.20 | 4.5 ± 6.2 |
| R49A | 0.0090 ± 0.0014 | 0.12 ± 0.16 | 17 ± 23 |
| E63A | 0.018 ± 0.00026 | 0.030 ± 0.0013 | 1.7 ± 0.074 |

*The affinity of YeeD C17A derivative for YeeE was too low, and the value could not be calculated accurately.

**The YeeD D42A sample contained a significant amount of intermolecular S-S dimers (S2F Fig); therefore, the value was omitted here.

### MD simulation of YeeE/YeeD complex

In the presence of thiosulfate ions, *St*YeeD was easily modified (Fig 2G) and the binding between *St*YeeE and *St*YeeD decreased (S4A and S4B Fig), leading to the hypothesis that the modified *St*YeeD dissociates from *St*YeeE. Here, to address how the *St*YeeE-YeeD complex undergoes structural changes when *St*YeeD is in the YeeD-SH or YeeD-S-$S_2O_3^-$ states, we conducted molecular dynamics (MD) simulation analysis (S5 Fig). In the case of *St*YeeD-SH, no significant structural changes were observed during the 1-μs MD simulation (S5A Fig): The simulation structure remained similar to the crystal structure of the *St*YeeE-YeeD complex (PDB ID 8K1R) (S5C Fig). On the other hand, in the case of *St*YeeD-S-$S_2O_3^-$, the *St*YeeD RMSD for Cα atoms increased from around 400 ns; in other words, *St*YeeD moved away from the cytoplasmic concave surface of *St*YeeE (S5A–S5C Fig). The calculation results are consistent with the idea that the interaction between YeeE and YeeD weakens upon Cys-modifications of YeeD. The dissociation of YeeD from YeeE may facilitate the release of thiosulfate ions' decomposition products from the Cys positions of YeeD into the cytoplasm.

### Discussion

By the growth complementation assay, we have demonstrated that YeeD is a required enzyme for growth when thiosulfate is solely imported via YeeE. YeeD's substrate was identified as the thiosulfate ion, and the uptake activity for thiosulfate ions was measured using YeeE-reconstituted liposomes. In addition, the interaction between YeeE and YeeD was revealed by both binding assay and X-ray crystallography. By mutational analysis of YeeD, a conserved cysteine residue was found to be critical for YeeE's activity in vivo as well as thiosulfate ion decomposition activity.

Compared with other TusA family proteins, YeeD shows some similarities and differences. Although conserved cysteine residues in other TusA family proteins, corresponding to C17 in *St*YeeD, are important for their own activities [10,20], TusA could not decompose thiosulfate ions [24]. *Metallosphaera cuprina* TusA was reported to exist as both monomer and dimer forms [12]. Similarly, *St*YeeD appeared to be monomers and dimers in gel filtration (S2C Fig).

*St*YeeD may exploit this feature to transiently form a dimer in the thiosulfate decomposition reaction (discussed below). The conserved cysteine residues in YeeD and TusA are important for binding to their respective partners (Fig 4) [12,18], which seems to be a common feature in TusA family proteins. Meanwhile, YeeD is unique in that its interaction partner, YeeE, is a membrane protein, unlike other TusA family proteins. Previously, the conserved cysteine residue of PA1006 was reported to be persulfide-modified [20]. Furthermore, *St*YeeD has several different sulfur-related modification statuses, including persulfide modification, perthiosulfonic acid, and thiosulfonate (Fig 2D and 2F–2H). In the case of DsrE3 and TusA, the conserved cysteines can possess thiosulfonate, but the modifications were detected only with their substrate tetrathionate [12]. Along with the fact that YeeE only allows thiosulfate to pass through, YeeD also seems to be specialized for thiosulfate reactivity. Both can contribute to the efficient uptake of thiosulfate. Since YeeE alone can transport thiosulfate ions but was unable to rescue the growth deficiency in vivo, cooperation between YeeE and YeeD is crucial. In this study, we have unveiled a sophisticated pathway for thiosulfate uptake in the bacterial membrane. An open question is whether there are other proteins that directly receive sulfur compounds from YeeD in sulfur assimilation pathways.

Based on our findings, we propose detailed mechanisms for thiosulfate ion uptake by YeeE and YeeD (S6A Fig). From the previous crystal structure of *St*YeeE, the transportation of thiosulfate ions across the membrane was proposed to be relayed by the first-to-third cysteine residues [6]. Our crystal structure of the *St*YeeE and *St*YeeD complex now further shows that YeeD is positioned to interact with the YeeE cytoplasmic side, which we propose to be the exit of the thiosulfate pathway on YeeE; the conserved cysteine residue of YeeD can function as a "fourth cysteine." Therefore, the interactions between the two can facilitate the delivery of a thiosulfate ion from YeeE to YeeD: The conserved cysteine residue of YeeD can directly capture a thiosulfate ion passed through YeeE. This transport scheme of thiosulfate ions by YeeE and YeeD recalls a group-translocation originally proposed by Mitchell and Moyle [25]. From our X-ray crystal structure, YeeE and YeeD appear to work as a 1:1 complex. The fact that YeeE and YeeD are encoded as a single polypeptide in some species supports this idea. After receiving the thiosulfate ion, YeeD's conserved cysteine would be thiosulfonated. Our BLI method analysis in the presence of thiosulfate showed that the interaction between *St*YeeD and *St*YeeE was weakened, possibly due to thiosulfonation on YeeD. The thiosulfonated YeeD may therefore dissociate from YeeE and be released into the cytoplasm. This dissociation model agrees well with our MD simulation results. Since YeeD can decompose thiosulfate by itself, it would do so after dissociating from YeeE. Of additional note, since the existence of *St*YeeD dimers in solution was suggested by gel filtration chromatography, *St*YeeD may form a transient dimer after being released from *St*YeeE.

We suggest a plausible mechanism for decomposition of thiosulfate ion by YeeD (S6B Fig). While the reducing power for this series of reactions is thought to be supplied from the cytoplasm, a reducing environment, the electron route remains unknown and needs to be elucidated in a future study. In the presence of thiosulfate ion, *St*YeeD incorporated thiosulfate, but *St*YeeD(C17A) did not (Fig 2E and 2G), suggesting that the thiosulfate ion directly binds to the cysteine residue of YeeD. Our MS results also suggest that there are *St*YeeD molecules with perthiosulfonic acid (-S-$SO_3^-$) and persulfide modifications (-S-$S^-$). Taking this together with the detection of $H_2S$ in the presence of *St*YeeD and thiosulfate ion (Fig 2H), it would be reasonable for sulfide ion ($S^{2-}$) to be released from a persulfide-modified version of *St*YeeD (-S-$S^-$) (S6B Fig). Although sulfonic acid (-$SO_3^-$) formation by the oxidization of a cysteine residue is an irreversible process, perthiosulfonic oxidization of a cysteine residue (-S-$SO_3^-$) is reversible as shown by Doka and colleagues [19]. Therefore, even if -S-$SO_3^-$ is formed, sulfite ion ($SO_3^{2-}$) will be released. In our experimental condition for the thiosulfate decomposition assay,

reductant β-ME is present to facilitate these reactions. In the cell, there could be other acceptor proteins that take S or $SO_3$ atoms from YeeD protein. Such proteins should be identified in a future study. While the significance of the *St*YeeD dimer in this study has not been demonstrated, if it were involved in this series of reactions, the scheme depicted in the S6B Fig box would likely occur. Because *St*YeeD does not form disulfide bonds in its dimer (S2C and S2D Fig), 1 StYeeD molecule in a dimer can be thiosulfonated and the other can have an active -SH group, which can take the $SO_3$ from thiosulfonated YeeD ($-S\text{-}S_2O_3^-$). Through this process, a persulfide-modified form ($-S\text{-}S^-$) can be generated. In some bacteria, including *E. coli*, YeeD has 2 cysteine residues. In species having YeeD with 2 cysteine residues, the abovementioned process could be carried out by 1 YeeD molecule using these 2 cysteine residues. Another possibility is that the nonconserved cysteine (C39) of *Ec*YeeD may not serve as the decomposition active center but may interact with YeeE and/or enhance its activity. In agreement with these ideas, our growth complementation assay showed that both cysteines are important for the growth of *E. coli* cells.

In the 2.60-Å structure of *St*YeeE-*St*YeeD, the electron density is faint on the N-terminal side of the CPxP motif, raising the possibility that some electrons may be attracted to the CPxP motif (Fig 3D). If that happens, the CPxP motif could make the region electron-rich and exhibit high reducing power. Previously, the CPxP architecture was proposed to stabilize an α-helix-1 by capping it [17]. In the TusA family proteins, the formation of CPxP may also provide high reducing power, conferring on them the unique property of selective decomposition of their substrates.

In conclusion, we revealed that YeeE and YeeD cooperatively contribute to the thiosulfate uptake pathway and possess unprecedented regulation mechanisms. Our findings also deepen the understanding of the functions of TusA family proteins.

## Materials and methods

### Strains and plasmids

For the growth complementation assay, we used plasmids derived from pAZ061, which is based on pET-16b-TEV [26] and possesses *E. coli* *yeeE* and *yeeD* genes between BamHI and XhoI sites amplified from *E. coli* genomic DNA (JCM 20135, RIKEN BRC) using primer set 5′-AAATTTATATTTTCAAGGATCCCATATGTTTTCAATGATATTAAG-3′ and 5′-GGCTTTGTTAGCAGCCCTCGAGTCAGGCTTTTTTGAACGG-3′ (S2A Fig), and the *E. coli* MG1655 Δ*cysPUWA* Δ*yeeE*::*kan* (DE3) strain, which cannot grow on a minimum medium with thiosulfate ion as the single sulfur source [6]. pAZ061 derivatives having mutations in the *yeeD* region were prepared by site-directed mutagenesis. The DNA sequence encoding *Ec*YeeD was deleted from pAZ061 using primers 5′-GTTTGGGTTAGGCATCGCTTCCC CAACGGCC-3′ and 5′-GGCCGTTGGGGAAGCGATGCCTAACCCAAAC-3′.

The plasmid used to express *St*YeeE (1-328aa, UniProt ID: G0GAP6) with C-terminal GSSGENLYFQGEDVE-His$_8$ sequence (pKK550) was prepared as in [6]. For the expression of the *St*YeeE-*St*YeeD fusion protein, we modified pKK550; the resulting plasmid, pAZ150, encodes *St*YeeE (1–330 aa)-AATPTPVAEAAPSSAEDRVLPFQVATGAVALQTAPRVKKA-*St*YeeD (1–80 aa, UniProt ID: G0GAP7)-GSSGENLYFQGEDVE-His$_6$. The mutations in the *St*YeeE and *St*YeeD regions were introduced by site-directed mutagenesis.

For the expression of *St*YeeD, the DNA sequence encoding *St*YeeD (1–80 aa) was inserted into a modified pCGFP-BC plasmid. The resulting plasmid, pNT015, expresses *St*YeeD (1–80 aa)-GSSGENLYFQGEDVE-His$_6$. The pNT015 derivatives were prepared by Gene Synthesis and Mutagenesis (SC1441, GenScript) or site-directed mutagenesis.

## Growth complementation tests of *E. coli* cells

*E. coli* MG1655 *ΔcysPUWA ΔyeeE::kan* (DE3) cells harboring pAZ061 or its derivatives were cultured in LB medium containing 50 μg/ml ampicillin and 25 μg/ml kanamycin for 16 h at 37˚C. The culture was diluted 100-fold and cultured for another 8 h at 37˚C, after which the cells were precipitated by centrifugation and washed twice with S-free medium (42 mM $Na_2HPO_4$, 22 mM $KH_2PO_4$, 8.6 mM NaCl, 19 mM $NH_4Cl$, 1 mM $MgCl_2$, 0.2% (w/v) glucose, 0.01% (w/v) thiamine hydrochloride, and 0.1 mM $CaCl_2$). Subsequent passage cultures in S-free medium containing 500 μM $Na_2S_2O_3$ were started at $OD_{600}$ = 0.2. The $\Delta OD_{600}$ of the culture was measured every 30 min with OD-Monitor C&T (TAITEC) for 24 h. Measurement was performed 3 times for each transformant.

## Protein expression and purification

*St*YeeE and *St*YeeE(C22A)-*St*YeeD proteins were purified as follows. *E. coli* C41(DE3) cells transformed with plasmids expressing these proteins were cultured in 2.5 l of LB medium containing 50 μg/ml ampicillin until the $OD_{600}$ reached 0.4. Isopropyl β-D-thiogalactopyranoside (IPTG) was then added to 1 mM, and the cells were cultured at 30˚C for 17 h. Subsequent procedures were performed at 4˚C or on ice. The cells were collected by centrifugation as pellets, suspended in a buffer (10 mM Tris-HCl (pH 8.0), 300 mM NaCl, 1 mM EDTA-Na (pH 8.0), 2 mM $Na_2S_2O_3$, 0.1 mM phenylmethylsulfonyl fluoride (PMSF), and 1 mM β–mercaptoethanol (β-ME)), and disrupted with a Microfluidizer Processor M-110EH (Microfluidics International). The suspension was centrifuged (10,000 rpm for 20 min, himac R13A rotor), and the collected supernatant was further ultracentrifuged (40,000 rpm for 60 min, Beckman 45Ti rotor) to obtain the membrane fraction, which was flash-frozen in liquid nitrogen and stored at −80˚C until purification. The membrane fraction was resuspended in solubilization buffer (20 mM Tris-HCl (pH 8.0), 300 mM NaCl, 2 mM $Na_2S_2O_3$, 10 mM imidazole-HCl (pH 8.0), 5% (v/v) glycerol, 1 mM β-ME, 0.1 mM PMSF) containing 1% (w/v) n-dodecyl β-maltoside (DDM) and stirred at 4˚C for 60 min. The insoluble fraction was removed by ultracentrifugation (45,000 rpm for 30 min, Beckman 70Ti rotor), and the supernatant was mixed with 5 ml of Ni-NTA agarose (QIAGEN) pre-equilibrated with solubilization buffer for 60 min. Then, 100 ml of wash buffer (20 mM Tris-HCl (pH 8.0), 300 mM NaCl, 2 mM $Na_2S_2O_3$, 50 mM imidazole-HCl (pH 8.0), 5% (v/v) glycerol, 1 mM β-ME, 0.1 mM PMSF, 0.1% (w/v) DDM) was added to the column. Next, 5 ml of elution buffer (20 mM Tris-HCl (pH 8.0), 300 mM NaCl, 2 mM $Na_2S_2O_3$, 200 mM imidazole-HCl (pH 8.0), 5% (v/v) glycerol, 1 mM β-ME, 0.1 mM PMSF, and 0.1% (w/v) DDM) was added to the column 6 times, and the fractions with target proteins were pooled. To remove the His-tag, TEV(S219V) protease [27] was added with a protein weight ratio of 10 (proteins):1 (TEV) and the protein solution was dialyzed overnight against dialysis buffer (20 mM Tris-HCl (pH 8.0), 300 mM NaCl, 2 mM $Na_2S_2O_3$, 0.1% (w/v) DDM, 1 mM β-ME). Ni-NTA agarose (2.5 ml) pre-equilibrated with dialysis buffer was added to the sample and stirred for 60 min to remove TEV proteases. The flow-through fraction was collected, concentrated with an Amicon Ultra 50K NMWL (Merck Millipore), and ultracentrifuged (45,000 rpm for 30 min, himac S55A2 rotor). The sample was then applied to a Superdex 200 increase 10/300 GL column (Cytiva) equilibrated with a buffer (20 mM Tris-HCl (pH 8.0), 300 mM NaCl, 2 mM $Na_2S_2O_3$, 0.1% (w/v) DDM, 1 mM β-ME). The fractions with target proteins were collected and pooled. *St*YeeE(C22A)-*St*YeeD(L45A) protein was purified as for *St*YeeE and *St*YeeE(C22A)-YeeD except that the buffer lacked $Na_2S_2O_3$.

 *St*YeeD purification was performed as follows. *E. coli* C41(DE3) cells containing *St*YeeD-encoding plasmids were inoculated into 25 ml of LB medium containing 50 μg/ml ampicillin and cultured overnight at 37˚C. The culture was then added to 0.5 l of LB medium containing

50 µg/ml ampicillin and allowed to grow until $OD_{600}$ = 0.4. After the addition of IPTG to 1 mM, the cells were cultured at 30˚C for 17 h. The cells were collected by centrifugation (4,500 rpm for 10 min, himac R9A2), flash-frozen in liquid nitrogen, and stored at −80˚C until use. The next procedures were performed at 4˚C or on ice. The frozen pellets were suspended in 25 ml of sonication buffer (20 mM Tris-HCl (pH 8.0), 300 mM NaCl, 10 mM imidazole-HCl (pH 8.0), 1 mM β-ME, 0.1 mM PMSF), sonicated for 10 min on ice with a Q500 sonicator (QSONICA), and centrifuged (15,000 rpm for 30 min, himac R13A). Ni-NTA agarose (2.5 ml) pre-equilibrated with the sonication buffer was added to the supernatant and rotated for 1 h. After the flow-through fraction was removed, the resin was washed with 125 ml of a buffer (20 mM Tris-HCl (pH 8.0), 300 mM NaCl, 20 mM imidazole-HCl (pH 8.0), 1 mM β-ME, 0.1 mM PMSF). Next, 5 ml of a buffer (20 mM Tris-HCl (pH 8.0), 300 mM NaCl, 200 mM imidazole-HCl (pH 8.0), 1 mM β-ME, 0.1 mM PMSF) was added, and the elution fraction was collected. The elution step was repeated 6 times. The fractions with target proteins were pooled and concentrated using an Amicon Ultra 3K NMWL (Merck Millipore). The sample was ultracentrifuged (45,000 rpm × 30 min, himac S55A2 rotor) and loaded onto a Superdex 200 increase 10/300 GL column pre-equilibrated with YeeD gel filtration buffer (10 mM Tris-HCl (pH 8.0) and 300 mM NaCl). The fractions with StYeeD proteins were concentrated using an Amicon Ultra 3K NMWL.

## MALDI-TOF MS

For StYeeD(WT) sample preparation for MS (Fig 2D), StYeeD(WT) protein was mixed with gel filtration buffer, so that the final concentration of StYeeD(WT) was 48.2 µM and the total volume was 200 µl, and incubated for 2.5 h at 37˚C. The sample was then ultracentrifuged, and the buffer was exchanged with MS buffer (10 mM Tris-HCl (pH 8.0)) using a NAP-5 Column (Cytiva). The final concentration of StYeeD(WT) for MS analysis was 17.5 µM.

For StYeeD(C17A) sample preparation for MS (Fig 2E), 48.2 µM StYeeD(C17A) in gel filtration buffer was incubated with or without 500 µM thiosulfate for 2.5 h at 37˚C in a total volume of 200 µl. After the incubation, the sample was ultracentrifuged, and the buffer was exchanged with MS buffer using a NAP-5 Column. The final concentrations of StYeeD(C17A) were 12.3 µM for without thiosulfate and 16.6 µM for with thiosulfate.

For the DTT condition of StYeeD(WT) (Fig 2F), DTT was added to StYeeD(WT) to give final concentrations of 401 µM and 16.7 mM, respectively, and a total volume of 288 µl. The sample was then incubated for 10 min at 37˚C and ultracentrifuged. A 48-µl aliquot was taken and the buffer was exchanged with MS buffer using a NAP-5 Column. The concentration of resultant DTT-treated StYeeD(WT) was 10.96 µM. For the thiosulfate or hydrogen peroxide condition of StYeeD(WT), the DTT-treated StYeeD(WT) sample was processed as follows. First, to remove DTT, the sample was applied to a Superdex 200 Increase 10/300 GL column (Cytiva) pre-equilibrated with gel filtration buffer. Next, the sample after gel filtration was concentrated using Amicon ultra 3K NMWL. For the thiosulfate condition (Fig 2G), the sample was then mixed with thiosulfate so that the final concentrations of StYeeD(WT) and thiosulfate were 48.2 µM and 500 µM, respectively, and the total volume was 200 µl. After incubating for 2.5 h at 37˚C, the sample was ultracentrifuged, and the buffer was exchanged with MS buffer using a NAP-5 Column. The final concentration of StYeeD(WT) for MS analysis was 14.0 µM. For the hydrogen peroxide condition (Fig 2H), the sample after gel filtration and concentration was mixed with hydrogen peroxide so that the final concentrations of StYeeD(WT) and hydrogen peroxide were 48.2 µM and 0.1% (w/v), respectively, and the total volume was 200 µl. After incubating for 10 min at room temperature, the sample was ultracentrifuged, and the buffer was exchanged with MS buffer using a NAP-5 Column. The final concentration of StYeeD(WT) for MS analysis was 23.8 µM.

Samples for MALDI-TOF analysis were prepared by the sinapinic acid (SA) double layer method [28]. In brief, 1 μl matrix solution A (saturated solution of SA in ethanol) was deposited onto the MALDI target and allowed to dry. Matrix solution B (saturated solution of SA in TA30 solvent (1:2 (v/v) acetonitrile:0.1% (v/v) trifluoroacetic acid in water)) and protein solution were mixed at a ratio of 1:24 (v/v). Matrix solution B/protein mixture was then deposited onto the matrix spot and allowed to dry. The dried samples were analyzed by Autoflex-II (Bruker Daltonics) with a linear positive mode and acquisition mass range of 3,000 to 20,000 Da. Theoretical m/z values were calculated with the equation of m/z = (M + n) / n, where M is molecular mass and n is the charge (n = 1 was adopted).

## Measurement of enzyme activity of *St*YeeD

The release of $H_2S$ due to chemical decomposition of thiosulfate ions by *St*YeeD was monitored by HSip-1 (Dojindo, SB21-10). The reaction solution contained 19.6 μM HSip-1, 24.1 μM YeeD, and 70 μM β-ME in the YeeD gel filtration buffer. As negative controls, 24.1 μM BSA was used instead of YeeD, or no protein was added to the solution. The reaction was started by adding 5 μl of 100 mM $Na_2S_2O_3$ at a 1:200 [v/v] ratio (final, 500 μM $Na_2S_2O_3$), and the solution was incubated at 37°C. The fluorescence of HSip-1 was measured using a fluorescence spectrophotometer (F-7000, Hitachi), at an excitation wavelength of 491 nm and an emission wavelength of 521 nm, every 30 min until 150 min.

## Crystallization

Purified *St*YeeE(C22A)-*St*YeeD fusion protein was concentrated to 21.5 mg/ml by Amicon Ultra 50K NMWL and crystallized using the lipidic cubic phase (LCP) method, as previously performed in the case of purified *St*YeeE [6]. Fifteen μl of 21.5 mg/ml *St*YeeE(C22A)-*St*YeeD was mixed with 4.3 μl of a buffer (20 mM Tris-HCl (pH 8.0), 300 mM $Na_2S_2O_3$, 0.1% (w/v) DDM, 1 mM β-ME) and incubated on ice for 20 min. Then, 16 μl of the sample was mixed with 24 μl of monoolein (M-239, Funakoshi) in an LCP syringe (Art Robbins Instruments) for 10 min. After 30 min of incubation at 20°C, 30 nl of the mixed samples were spotted onto MRC under oil crystallization plates (Hampton Research) using the Crystal Gryphon protein crystallization aliquot system (Art Robbins Instruments) with 3 μl of buffers (18 to 24% (v/v) pentaerythritol-propoxylate (5/4 PO/OH), 100 mM 2-morpholinoethanesulfonic acid (MES)-NaOH (pH 7.0), and 300 mM NaCl) covering them. The plate was incubated at 20°C for 7 d. For *St*YeeE(C22A)-*St*YeeD(L45A) fusion protein, the purified protein was concentrated to 16.5 mg/ml and crystallized using the LCP method. The crystals appeared using 0.35 M ammonium formate, 0.1 M Tris-HCl (pH 8.0 to 9.0), and 22 to 42% (v/v) 1,4-Butanediol as covering solutions. The plate was incubated at 20°C for 14 d. The crystals that appeared were harvested by Crystal Mounts and Loops (MiTegen) without using cryoprotectant, directly flash-frozen in liquid nitrogen, and stored in liquid nitrogen until X-ray diffraction experiments.

## Data collection and structural determination

X-ray diffraction experiments of *St*YeeE(C22A)-*St*YeeD and *St*YeeE(C22A)-*St*YeeD(L45A) were performed at beamline BL32XU of SPring-8. Data were collected with the automated data collection system ZOO [29]. The complete data sets were obtained by combining multiple small wedge datasets from hundreds of tiny crystals with sizes of approximately 20 μm. The principle of this method has been described in detail [30]. Data processing was performed with KAMO [30] using XDS [31] programs. For *St*YeeE(C22A)-*St*YeeD dataset, the initial phase was determined by molecular replacement using the *St*YeeE crystal structure (PDB ID: 6LEO) as a template by PHASER [32], and a *St*YeeD structure predicted by AlphaFold2 [15,16] was

manually fitted to the density map using COOT [33]. Structure refinement was performed using COOT [33] and PHENIX [34] iteratively until $R_{work}/R_{free}$ reached 0.265/0.310 at 3.34 Å resolution. For $St$YeeE(C22A)-$St$YeeD(L45A) dataset, the initial phase was determined by molecular replacement using the $St$YeeE(C22A)-$St$YeeD structure as a template by MOLREP [35]. Refinement of the structure was performed using COOT [33] and PHENIX [34] in a iterative way until $R_{work}/R_{free}$ reached 0.206/0.255 at 2.60 Å resolution. Figures of the structures were prepared using PyMOL (https://pymol.org/2/) and Chimera [36].

## Interaction analysis between $St$YeeE and $St$YeeD by the BLI method

BLI method [23] was performed to analyze the interaction between $St$YeeE and $St$YeeD-His$_6$ proteins using the Octet N1 System (Sartorius) at room temperature. A Ni-NTA biosensor was hydrated with Octet buffer (20 mM Tris-HCl (pH 8.0), 300 mM NaCl, 0.1% (w/v) DDM, 1 mM β-ME) for 10 min and mounted in the Octet N1 System. The biosensor was first dipped in Octet buffer, and the initial baseline was measured for 30 s. Next, the biosensor was dipped in 4.82 μM $St$YeeD-His$_6$ protein solution in Octet buffer for 120 s for loading. The solution was then changed to Octet buffer for 30 s to measure the baseline, after which, the biosensor was submerged in 11.05 μM $St$YeeE solution in Octet buffer for 120 s to measure the association of $St$YeeE. Finally, the buffer was exchanged to Octet buffer, and the dissociation of $St$YeeE from the biosensor was measured for 120 s. Biosensors without $St$YeeD-His$_6$ proteins were measured with the same procedure and used as references. After the reference data were subtracted from $St$YeeD-His$_6$ protein data, the association rate constant ($k_a$), dissociation rate constant ($k_d$), and affinity constant ($K_D$) were calculated by a local fitting method. Measurements were performed 3 times for each protein.

## Reconstitution of proteoliposomes

A mixture of 0.8 mg/ml $St$YeeE and 4 mg/ml $E. coli$ total lipid extract (Avanti Polar Lipids) in a buffer (20 mM HEPES-HCl (pH 7.0), 300 mM NaCl, 5% (v/v) glycerol, and 0.1% (w/v) DDM) was rotated at 4°C for 1 h. Next, the detergent was removed using SM2-beads (Bio-Rad). The resulting solution was ultracentrifuged, and the precipitates were suspended in a buffer (25 mM HEPES-NaOH (pH 7.5) and 100 mM NaCl). The reconstituted proteoliposome samples were flash-frozen in liquid nitrogen and stored at −80°C until measurements.

## Measurement of ion transport activity

The SSM method was used to detect thiosulfate uptake activity of the $St$YeeE-reconstituted proteoliposomes. Frozen proteoliposomes were thawed on ice and sonicated using a bath sonicator (VELVO-CLEAR, VS-50R) for 10 s 3 times before use. Measurement was performed using SURFE$^2$R N1 (Nanion Technologies) as described [14]. First, 50 μl of 0.5 mM 1-octadecanethiol (dissolved in isopropanol) was applied to N1 Single Sensors (Nanion Technologies, Nr. 2-03-35002-000) and incubated for 30 min at room temperature. Next, the sensors were washed twice with isopropanol, washed twice with distilled water, and dried. Then, 1.5 μl of 7.5 μg/μl 1,2-diphytanoyl-sn-glycero-3-phosphocholin (dissolved in n-decane) was applied to the sensor, followed by 50 μl of nonactivating buffer (B buffer: 140 mM NaCl, 4 mM MgCl$_2$, 25 mM HEPES, 25 mM MES, KOH (pH 6.7), with or without Na$_2$SO$_4$). Proteoliposomes were pipetted toward the sensor beneath the B buffer surface, and the sensor was centrifuged (2,000$g$ for 30 min, 25°C) to adsorb the liposomes onto the sensor surface. The resultant sensors were mounted in SURFE$^2$R N1, and the sensors were rinsed with buffer B before each measurement. The current change on the sensor was monitored while B buffer and A buffer (140 mM NaCl, 4 mM MgCl$_2$, 25 mM HEPES, 25 mM MES, KOH (pH 6.7), with Na$_2$S$_2$O$_3$)

were exchanged. For each condition, measurement was performed four times, and the 3 results with least noise were analyzed.

## MD simulations of *St*YeeE-*St*YeeD complex

All-atom MD simulations of the StYeeE-StYeeD complex were performed in POPC lipid bilayers and 150 mM NaCl solution, where 2 intermediate states, *St*YeeE-*St*YeeD-SH and *St*YeeE-*St*YeeD-S-$S_2O_3^-$, were examined. The initial structure of the protein-membrane complex was constructed using the CHARMM-GUI *membrane builder* [37]. The system size is 100 Å × 100 Å × 120 Å, and the total number of atoms is approximately 111,600. The CHARMM C36m force field parameters were used for proteins and lipids [38], and the topology and parameters for S-$S_2O_3^-$-modified Cys were derived from the CGenFF parameters [39]. The system was equilibrated using 2,000-step energy minimization, followed by equilibration in the *NVT* and *NPT* ensembles for 3.5 ns using the positional restraints on the proteins and lipids. The production run was then carried out for 1 μs in the *NPT* ensemble at $T = 310$ K and $P = 1$ atm with the time step of 3.5 fs using the RESPA integrator with hydrogen-mass repartitioning and bond-length constraining [40] and the Bussi thermostat and barostat for temperature and pressure control [41]. The particle-mesh Ewald method was used for the calculation of long-range electrostatic interaction [42]. All simulations were performed using GENESIS [43,44].

## Supporting information

**S1 Fig. Sequence alignment of YeeDs from 13 species.** The UniProt IDs of YeeD sequences used are as follows: *E. coli* YeeD, P33014; *Spirochaeta_thermophila* YeeD, G0GAP7; *Candidatus Sodalis pierantonius* YeeD, W0HL40; *Klebsiella_sp._WP3-W18-ESBL-02* YeeD, A0A7I6Q8F8; *Methanosarcinales archaeon* YeeD, A0A822J3Z6, *Streptococcus thermophilus* YeeD, A0A8D6XUG1; *Koleobacter methoxysyntrophicus* YeeD, A0A8A0RNL6; *Wohlfahrtiimonas chitiniclastica* YeeD, L8Y0N6; *Paenibacillus alvei* YeeD, A0A383RJV0; *Shigella flexneri* YeeD, A0A384L8W9; *Streptococcus gallolyticus* YeeD, A0A380K504; *Micrococcus_sp._116* YeeD, A0A653IT90; *Magnetospirillum gryphiswaldense* YeeD, V6F4H7. The red rectangle indicates the CPxP motif. The first and second cysteine residues (solid and open red arrowheads) are completely and not completely conserved among species, respectively. The figure was generated using ESPript 3.0 [45].
(TIFF)

**S2 Fig. Data related to functional analysis of YeeD.** (**A**) Details of plasmid (pAZ061), used as the positive control for the growth complementation assay. Tandemly located *EcyeeE* and *EcyeeD* are regulated by the same promoter. A His$_{10}$-tag is attached to the N-terminal side of *Ec*YeeE. Based on pAZ061, a deletion and several point mutations on *EcyeeD* were introduced. (**B**) Original data from the growth complementation assay in Fig 2C. Error bars represent the SD of 3 measurements. (**C**) Gel filtration profile of purified *St*YeeD(WT), which eluted with 2 peaks (red arrowheads). The eluted positions of standard proteins and their molecular masses are shown. (**D**) Nonreducing (−βME) and reducing (+βME) SDS-PAGE of *St*YeeD after iodoacetamide (IAA)- or N-Ethylmaleimide (NEM) treatment. Only minor fractions show intermolecular disulfide bond formation between *St*YeeDs(WT). (**E**) Irreversible oxidation of *St*YeeD by hydrogen peroxide. Before reducing SDS-PAGE, the hydrogen peroxide treatment was performed. (**F**) Nonreducing SDS-PAGE profile of *St*YeeD mutants after NEM treatment. (**G**) Original data for the enzymatic activity of *St*YeeD detected by HSip-1 in Fig 2I. Error bars show the SD from 3 measurements. The underlying data for (B), (C), and (G) can be found in

S1 Data. The original gel images of (D–F) can be found in S1 Raw Images.
(TIFF)

**S3 Fig. Structural comparisons of *St*YeeE-YeeD complex.** (**A**) A comparison of the crystal structures Mol A and Mol B in the asymmetric unit of 8J4C. (**B**) Crystal packing of 8J4C viewed from 2 different directions. (**C**) Comparison of the crystal structures of Mol A of 8K1R and MolA of 8J4C. (**D and E**) Comparisons of the crystal structures of *St*YeeE-YeeD complex (MolA) and *St*YeeE (PDB ID 6LEP) (D) or *E. coli* TusA (PDB ID 3LVK) (E).
(TIFF)

**S4 Fig. Data related to interactions between YeeE and YeeD.** (**A**) Real-time detection by the BLI method of association and dissociation of *St*YeeE with/from solidified *St*YeeD in buffer without a sulfur source, with 300 mM $Na_2SO_4$, or with 300 mM $Na_2S_2O_3$. Each line shows the mean value of 3 measurements with the SD. The dashed line indicates the 120-s point. (**B**) Comparison of mean values of *St*YeeE binding to solidified *St*YeeD at 120 s. Error bars represent the SD of 3 measurements. Statistical significance compared with no sulfur source was determined using one-way analysis of variance (ANOVA) followed by Dunnett's multiple comparisons tests (\*\*, $p < 0.01$; ns, not significant). (**C**) Growth complementation assay of $\Delta cysPUWA\Delta yeeE$ (DE3) cells, depending on *Ec*YeeE and *Ec*YeeD expressed from plasmids. Mean values of increased $OD_{600}$ ($\Delta OD_{600}$) after 24 h are shown. Error bars represent the SD from 3 measurements. Statistical significance compared with the $\Delta cysPUWA \Delta yeeE$ (DE3) cells possessing an empty vector was determined using one-way ANOVA followed by Dunnett's multiple comparisons tests (\*\*\*\*, $p < 0.0001$; \*\*, $p < 0.01$; ns, not significant). The underlying data for (A–C) can be found in S1 Data. (**D**) Mutation site mapping on the crystal structure of *St*YeeE-YeeD complex (PDB ID 8J4C). The side chains of amino acid residues of *St*YeeE corresponding to those of *Ec*YeeE mutated in (C) are shown as stick models with the same colors as in (C). The amino acid residues of *St*YeeD shown in Fig 4C are also indicated.
(TIFF)

**S5 Fig. All-atom MD simulations of *St*YeeE/YeeD-SH and *St*YeeE/YeeD-S-$S_2O_3^-$.** (**A**) Time courses of the Cα-RMSD calculated for *St*YeeD regions relative to the initial structures. For the RMSD calculations, each *St*YeeE region was superimposed. (**B**) Snapshots in the MD simulation of *St*YeeE/YeeD-S-$S_2O_3^-$. (**C**) Comparison between the initial structure (0 ns) and final snapshot (1,000 ns) for *St*YeeE/YeeD-SH (left) and *St*YeeE/YeeD-S-$S_2O_3^-$ (right). The square boxes display magnified views of the region around C17 (spheres). The underlying data can be found in S2 Data.
(TIFF)

**S6 Fig. Model of uptake and decomposition of thiosulfate ion by YeeE and YeeD.** (**A**) Uptake of a thiosulfate ion by YeeE and YeeD. The thiosulfate ion is relayed by conserved cysteine residues of YeeE and binds to the conserved cysteine residue of YeeD. After that, the association of YeeD with YeeE is destabilized. YeeD then dissociates from YeeE and decomposes the thiosulfate. (**B**) Thiosulfate decomposition by YeeD. First, thiosulfate binds directly to the conserved cysteine residue of YeeD. Next, sulfide ion ($SO_3^{2-}$) is released. Finally, sulfite ion ($S^{2-}$) is released. In the blue box, a putative dimerization model of *St*YeeD is shown.
(TIFF)

**S1 Data. Underlying data.** Raw data for Figs 1A–1C, 2C–2I, 4A, and 4B, S2B, S2C, S2G, and S4A–S4C.
(XLSX)

**S2 Data. MD simulation results.** The initial models (PDB file format) and the MD simulation results (DCD file format).
(ZIP)

**S1 Raw Images. The untrimmed original TIFF images for Figs 2D, 2E, and S2D–S2F.**
(ZIP)

# Acknowledgments

We thank Kayo Abe for secretarial assistance, Akira Sasaki for liposome preparation, Rie Kurata for performing mass spectrometry, Naoki Sakai for helping with the data analysis, the beamline scientists at BL32XU of SPring-8 (Hyogo, Japan) for helping with data collection, and Nanion Technologies for assistance with initial data collection. The synchrotron radiation experiments were performed at BL32XU of SPring-8 with the approval of the Japan Synchrotron Radiation Research Institute (JASRI) (Proposal Nos. 2020A2564, 2021A2745, 2022A2738, 2023A2727).

# Author Contributions

**Conceptualization:** Azusa Takeuchi, Muneyoshi Ichikawa, Tomoya Tsukazaki.

**Investigation:** Mai Ikei, Ryoji Miyazaki, Keigo Monden, Yusuke Naito, Azusa Takeuchi, Yutaro S. Takahashi, Yoshiki Tanaka, Keina Murata, Takaharu Mori, Muneyoshi Ichikawa, Tomoya Tsukazaki.

**Methodology:** Mai Ikei, Ryoji Miyazaki, Keigo Monden, Yusuke Naito, Azusa Takeuchi, Yutaro S. Takahashi, Yoshiki Tanaka, Keina Murata, Takaharu Mori, Muneyoshi Ichikawa, Tomoya Tsukazaki.

**Supervision:** Muneyoshi Ichikawa, Tomoya Tsukazaki.

**Writing – original draft:** Muneyoshi Ichikawa, Tomoya Tsukazaki.

**Writing – review & editing:** Muneyoshi Ichikawa, Tomoya Tsukazaki.

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
