## [Editor Report · Decision Letter 0]

12 Feb 2024

Dear Dr Tsukazaki, 

Thank you for submitting your revised manuscript from Review Commons entitled "Structure and function of bacterial YeeE-YeeD complex in thiosulfate uptake pathway" for consideration as a Research Article by PLOS Biology. Please accept my apologies for the long delay in getting back to you as we consulted with an academic editor about your submission. 

Your manuscript has now been evaluated by the PLOS Biology editorial staff, as well as by an academic editor with relevant expertise, and I am writing to let you know that we would like to send your manuscript back out for review by the original reviewers at Review Commons.

*IMPORTANT* - After discussions with the Academic Editor and within the editorial team, we would like to consider your manuscript as a Short Report (https://journals.plos.org/plosbiology/s/what-we-publish#loc-short-reports). During resubmission (see below), we would be grateful if you could tick 'Short Report' as the article type in the online submission form, as well as reducing the number of main figures to 4. This could be achieved by possibly moving the uptake model presented in Figure 5 to the Supplementary figures.

However, before we can send your manuscript to the original reviewers at Review Commons, we need you to complete your submission by providing the metadata that is required for full assessment. To this end, please login to Editorial Manager where you will find the paper in the 'Submissions Needing Revisions' folder on your homepage. Please click 'Revise Submission' from the Action Links and complete all additional questions in the submission questionnaire.

Once your full submission is complete, your paper will undergo a series of checks in preparation for peer review. After your manuscript has passed the checks it will be sent out for review. To provide the metadata for your submission, please Login to Editorial Manager (https://www.editorialmanager.com/pbiology) within two working days, i.e. by Feb 14 2024 11:59PM.

Kind regards,

Richard

Richard Hodge, PhD

rhodge@plos.org

PLOS

---

## [Decision Letter · Decision Letter 1]

12 Mar 2024

Dear Dr Tsukazaki,

Thank you for your patience while we considered your revised manuscript "Structure and function of bacterial YeeE-YeeD complex in thiosulfate uptake pathway" for publication as a Short Report at PLOS Biology. The revised version of your manuscript has been evaluated by the PLOS Biology editors, the Academic Editor and two of the original reviewers at Review Commons.

Based on the reviews, I am pleased to say that we are likely to accept this manuscript for publication, provided you satisfactorily address the following data and other policy-related requests that I have provided below (A-H):

(A) We would like to suggest the following modification to the title:

"YeeD is an essential partner for YeeE-mediated thiosulfate uptake in bacteria and regulates thiosulfate ion decomposition”

(B) You may be aware of the PLOS Data Policy, which requires that all data be made available without restriction: http://journals.plos.org/plosbiology/s/data-availability. For more information, please also see this editorial: http://dx.doi.org/10.1371/journal.pbio.1001797

-Supplementary files (e.g., excel). Please ensure that all data files are uploaded as 'Supporting Information' and are invariably referred to (in the manuscript, figure legends, and the Description field when uploading your files) using the following format verbatim: S1 Data, S2 Data, etc. Multiple panels of a single or even several figures can be included as multiple sheets in one excel file that is saved using exactly the following convention: S1_Data.xlsx (using an underscore).

-Deposition in a publicly available repository. Please also provide the accession code or a reviewer link so that we may view your data before publication. 

Figure 1A-C, 2C, 2I, 4A-B, S2B, S2G, S4A-C

(C) Please deposit the mass spectrometry data (Figure 2D-H) in a public data repository such as the ProteomeXchange (https://www.proteomexchange.org/). Please ensure that the data is made publicly available and provide the accession number in the Data Availability Statement in the online submission form. 

(D) Thank you for providing the structural data in the PDB database (8J4C and 8K1R). However, we note that the data is currently on hold for release. We ask that you please make the structures publicly available at this stage before publication.

(E) Please also ensure that each of the relevant figure legends in your manuscript include information on *WHERE THE UNDERLYING DATA CAN BE FOUND*, and ensure your supplemental data file/s has a legend.

(F) We require the original, uncropped and minimally adjusted images supporting all blot and gel results reported in the following Figures:

Figure S2D-F

We will require these files before a manuscript can be accepted so please prepare and upload them now. Please carefully read our guidelines for how to prepare and upload this data: https://journals.plos.org/plosbiology/s/figures#loc-blot-and-gel-reporting-requirements

(G) Please ensure that your Data Statement in the submission system accurately describes where your data can be found and is in final format, as it will be published as written there. 

(H) Please make any custom code available, either as a supplementary file or as part of your data deposition. Please ensure that any code that you have developed is sufficiently well documented and reusable, and that your Data Statement in the Editorial Manager submission system accurately describes where your code can be found.

We expect to receive your revised manuscript within two weeks. 

*Published Peer Review History*

*Press*

Sincerely,

Richard

Richard Hodge, PhD

rhodge@plos.org

Reviewer remarks:

Reviewer #2: All my comments were sufficiently revised and I recommend the acceptance of the article

The publication "Structure and function of the bacterial YeeE-YeeD complex in the thiosulfate uptake pathway" by Ikei et al. by Ikei et al. describes the protein-protein interaction and the structure of the thiosulfate importer YeeE and the sulfur transferase YeeD. The sulfur transferase YeeD is a protein of the TusA family, whose members are involved in diverse and highly variable sulfur transfer reactions, including dissimilatory sulfur oxidation, molybdenum cofactor assembly, and tRNA modification. The experiments by Ikei et al. complement previous descriptions of the YeeE transporter structure by providing a YeeE-YeeD crystal structure. The binding affinity between the two proteins and amino acid exchanged mutants were were measured by biolayer interferometry to determine the influence of specific surface-exposed of specific surface-exposed residues on the protein interaction. Mass spectrometry also showed the mobilization of free thiosulfate by YeeD. The interaction of transferase YeeD with the thiosulfate transporter YeeE is a novelty in the field. to the field. This is the first time that a specific function of YeeD in the thiosulfate assimilation. 

In a first evaluation a number of major and minor comments were made. These comments have been revised by the authors:

A number of major comments concerned experiments on the mobilization of free thiosulfate by the sulfur transferase YeeD. In these experiments, the formation of sulfinic acid was observed, which prevented clear conclusions about the degradation of thiosulfate. The reaction mechanism, electron flow, and resulting protein-bound sulfur species were unclear.

In the revised manuscript, the authors provided insight into the reaction mechanism by demonstrating mobilization of sulfur from free thiosulfate catalyzed by YeeD. For this effort, additional controls were performed and the sulfur mobilization experiments were completely repeated. The study of YeeD dimer formation and artificial oxidation by molecular oxygen was also extended, reducing the number of possible reaction mechanisms. The authors also made it clear that the artificial oxidation must be taken into account when interpreting the results. Taken together, these experiments justify the consideration of sulfur mobilization from thiosulfate by YeeD and support YeeD as an acceptor of thiosulfate from the importer YeeE. 

A second major comment concerned the proposed reaction mechanism for thiosulfate decomposition by YeeD, which was biochemically impossible. The authors revised this section and now propose a possible mechanism.

The exact mechanism of thiosulfate mobilization and the electron flow remains to be elucidated in future experiments. Considering the limited technical capabilities of the authors to perform experiments under anaerobic conditions, the comments were answered to my satisfaction.. 

The manuscript novelty is the structure and interaction of the TusA-family protein YeeD and the YeeE thiosulfate importer. The findings are relevant for a wide range of metabolisms and prokaryotes due to the wide distribution of YeeD and YeeE homologues and their involvement in various sulfur metabolism pathways. I recommend the acceptance of the article.

Reviewer #3: The manuscript "Structure and function of bacterial YeeE-YeeD complex in thiosulfate uptake pathway" by Ikei et al., reports the enzymatic characterization, transport capability and concerted function of YeeE and YeeD. Moreover, the authors report the crystal structures of two mutant variants of the complex. The present work fills an important gap in understanding thiosulfate uptake and the individual roles of the YeeE and YeeD proteins in this process. 

This Reviewer is reviewing this manuscript for the second time. This Reviewer congratulates the Authors on accepting so well the provided suggestions (and criticism), by the three Reviewers, and for incorporating all of them in the final versions of the manscript.

This Reviewer believes that the paper has the potential of becoming an important reference in the field. Especially now, after such an extensive and careful review,

---

## [Editor Report · Decision Letter 2]

26 Mar 2024

Dear Dr Tsukazaki,

On behalf of my colleagues and the Academic Editor, Lotte Søgaard-Andersen, I am pleased to say that we can accept your manuscript for publication, provided you address any remaining formatting and reporting issues. These will be detailed in an email you should receive within 2-3 business days from our colleagues in the journal operations team; no action is required from you until then. Please note that we will not be able to formally accept your manuscript and schedule it for publication until you have completed any requested changes.

PRESS

Kind regards, 

Richard

Richard Hodge, PhD

rhodge@plos.org

PLOS
